# Hypothalamic hormone deficiency enables physiological anorexia in ground squirrels during hibernation

Sarah M. Mohr [1,2,3], Rafael Dai Pra [1,2,3], Maryann P. Platt [1,2,3], Viktor V. Feketa[1,2,3], Marya Shanabrough [4], Luis Varela [4,5,6], Ashley Kristant[4], Haoran Cao[1,2,3], Dana K. Merriman [7], Tamas L. Horvath [4,5,6], Sviatoslav N. Bagriantsev [1] ✉ & Elena O. Gracheva [1,2,3] ✉

Mammalian hibernators survive prolonged periods of cold and resource scarcity by temporarily modulating normal physiological functions, but the mechanisms underlying these adaptations are poorly understood. The hibernation cycle of thirteen-lined ground squirrels *(Ictidomys tridecemlineatus)* lasts for 5–7 months and comprises weeks of hypometabolic, hypothermic torpor interspersed with 24–48-h periods of an active-like interbout arousal (IBA) state. We show that ground squirrels, who endure the entire hibernation season without food, have negligible hunger during IBAs. These squirrels exhibit reversible inhibition of the hypothalamic feeding center, such that hypothalamic arcuate nucleus neurons exhibit reduced sensitivity to the orexigenic and anorexigenic effects of ghrelin and leptin, respectively. However, hypothalamic infusion of thyroid hormone during an IBA is sufficient to rescue hibernation anorexia. Our results reveal that thyroid hormone deficiency underlies hibernation anorexia and demonstrate the functional flexibility of the hypothalamic feeding center.

In humans, inhibition of food intake, or anorexia, can manifest as a serious disorder that impacts quality of life and in severe cases can cause death. Anorexia can describe the lack of hunger due to pathological conditions, for example during cancer cachexia, or in the psychiatric condition anorexia nervosa, in which subjects self-limit eating despite a negative energy balance. Physiological anorexia is observed in toddlers and elderly adults who, during certain developmental periods, become less interested in food and reduce their food intake. The mechanisms underlying various types of anorexia remain poorly understood, though recent evidence strongly suggests that, in addition to psychological and social factors, anorexia is driven by physiological changes[1–6].

For thirteen-lined ground squirrels *(Ictidomys tridecemlineatus,* Fig. 1a), who do not rely on stored food during hibernation, anorexia is an essential component of a normal physiological cycle. Because premature emergence from the safety of the underground burrow to search for food would defeat the purpose of hibernation and pose a risk of predation, anorexia constitutes an important safety mechanism that increases survival.

A seasonal cycle of ground squirrels consists of several physiological states. In the active state (late spring to late summer), squirrels are euthermic, hyperphagic and metabolically active. During the pre-hibernation state (late summer to early fall), squirrels reduce food

[1]Department of Cellular and Molecular Physiology, Yale University School of Medicine, 333 Cedar Street, New Haven, CT 06510, USA. [2]Department of Neuroscience, Yale University School of Medicine, 333 Cedar Street, New Haven, CT 06510, USA. [3]Kavli Institute for Neuroscience, Yale University School of Medicine, 333 Cedar Street, New Haven, CT 06510, USA. [4]Department of Comparative Medicine, Yale University School of Medicine, 310 Cedar Street, New Haven, CT 06510, USA. [5]Laboratory of Glia-Neuron Interactions in the Control of Hunger. Achucarro_Basque Center for Neuroscience, 48940 Leioa, Vizcaya, Spain. [6]IKERBASQUE, Basque Foundation for Science, 48009 Bilbao, Vizcaya, Spain. [7]Department of Biology, University of Wisconsin-Oshkosh, 800 Algoma Boulevard, Oshkosh, WI 54901, USA. ✉e-mail: slav.bagriantsev@yale.edu; elena.gracheva@yale.edu

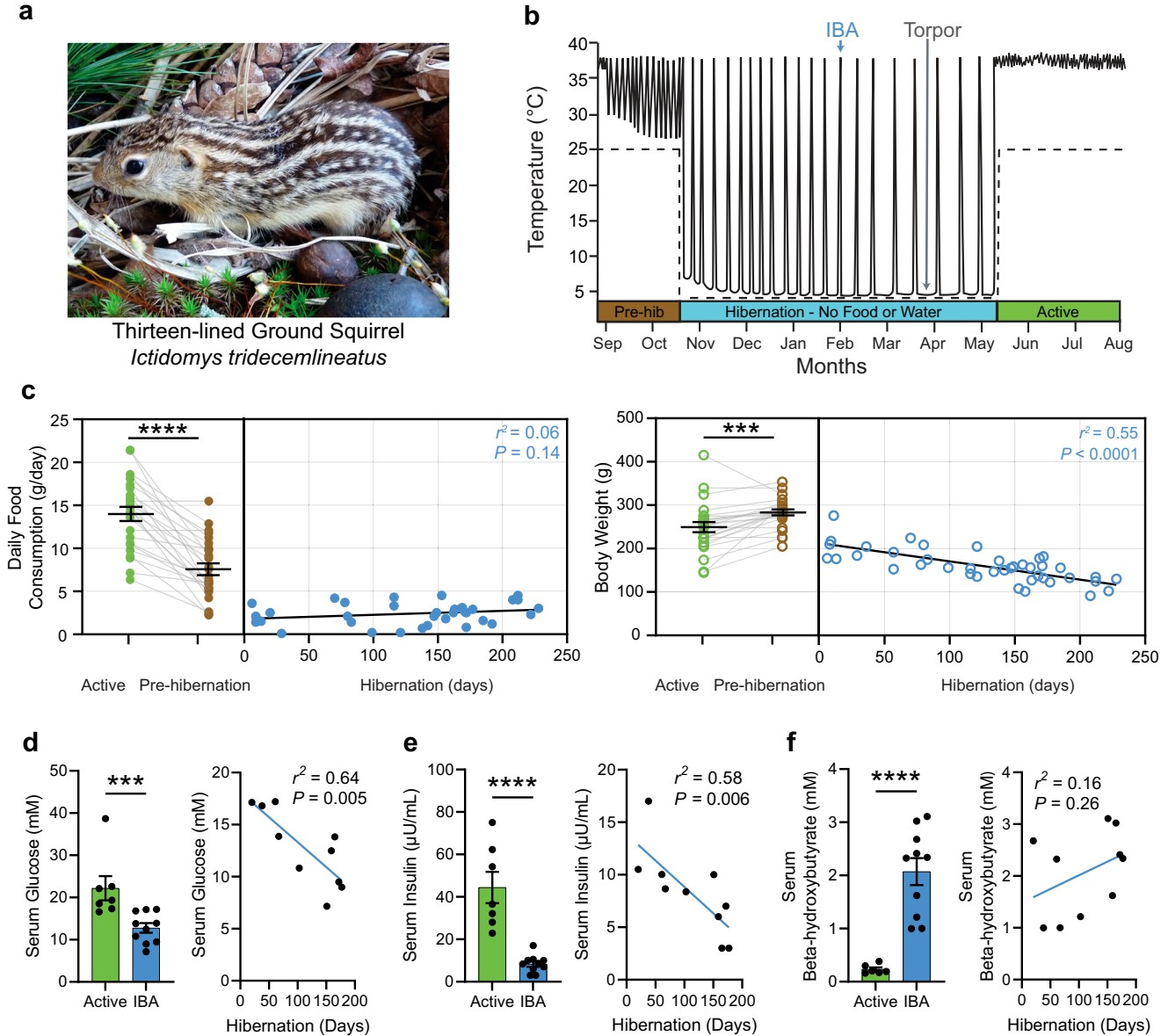

**Fig. 1 | Hibernating ground squirrels exhibit negligible food consumption.**
**a** Image of a thirteen-lined ground squirrel, *Ictidomys tridecemlineatus* (courtesy of the Gracheva laboratory). **b** Schematic of ground squirrel core body temperature before, during and after hibernation. Dotted line, ambient temperature. Every temperature peak during hibernation represents an IBA. *IBA*: interbout arousal, *Pre-hib*: Pre-hibernation. **c** Daily food consumption (left) and body weight (right) during the active (summer), pre-hibernation (fall), and hibernation (winter) season. Active feeding data is peak summer food consumption. Pre-hibernation feeding data is the last daily feeding measurement that the animal maintains euthermia (mean ± SEM, $n = 25$ Active and Pre-hibernation animals. Active and Pre-hibernation data points are paired. Feeding comparison: Two-sided student's paired *t*-test, $P = 0.006 \times 10^{-5}$; Body weight comparison: Two-sided student's paired *t*-test, $P = 0.0005$. Hibernation feeding

and body weight were measured during IBAs and are plotted by days in hibernation ($n = 36$ IBA animals for food consumption and $n = 42$ IBA animals for body weight, IBA data are independent biological replicates, simple linear regression). Blood metabolic indicators across active and hibernation states. Data across seasons are independent biological replicates and are represented as mean ± SEM in bar plots. Scatter plots are the same data as in the IBA bar plot, but plotted against days in hibernation and fitted with simple linear regressions. **d** Serum glucose ($n = 7$ Active and $n = 10$ IBA animals, Two-sided Mann-Whitney test, $P = 0.007$). **e** Serum insulin ($n = 7$ Active and $n = 10$ IBA animals, Two-sided student's *t*-test, $P = 0.00003$). **f** Serum beta-hydroxybutyrate ($n = 6$ Active and $n = 10$ IBA animals, Two-sided student's *t*-test, $P = 0.00007$). Each point represents one animal. ***$P < 0.001$, ****$P < 0.0001$. Source data are provided as a Source Data file.

consumption and undergo temporary bouts of hypothermia. Hibernation (early fall to late spring) consists of repeated cycles of hypothermic torpor interspersed with brief periods of euthermic IBA. During torpor, animals enter a state of suspended animation by profoundly reducing their metabolic, heart and respiration rates, and lowering their body temperature to 2–4 °C. Every 2–3 weeks, squirrels arouse to spend ~24 h in IBA (Fig. 1b), when their main bodily functions temporarily return to an active-like state[7]. Ground squirrels do not depend on stored food during hibernation; instead, energy is supplied

by body fat amassed during the summer. Thus, although hibernating squirrels resemble fasted animals metabolically, they demonstrate little interest in food despite enduring over seven months of starvation[8–11].

We sought to understand the mechanism underlying this remarkable example of reversible anorexia by comparing euthermic animals during the active season with euthermic animals during IBAs in the hibernation season. Our experiments show that infusing thyroid hormone into the mediobasal hypothalamus increases feeding during

IBA, strongly suggesting that hibernation anorexia is caused by thyroid hormone deficiency in the hypothalamus.

## Results

### Hibernating ground squirrels exhibit negligible food consumption

We found that when squirrels were presented with food during the numerous IBAs throughout the hibernation season, they consumed approximately six times less food than during the active state, and 3 times less than during the pre-hibernation state (Active: 14.0 ± 0.8 g/day, pre-hibernation: 7.6 ± 0.7 g/day, IBA: 2.4 ± 0.2 g/day; One-Way ANOVA with Dunnett's multiple comparison test: Active vs. Pre-hibernation, $P < 0.0001$; IBA vs. Active, $P < 0.0001$; IBA vs. Pre-hibernation, $P < 0.001$; Fig. 1c and Extended Data Fig. 1). Their body weight continuously decreased throughout hibernation, reaching ~50% of their starting weight by the end of the season (Fig. 1c and Extended Data Fig. 1). Concurrently with body weight reduction, IBA squirrels progressively reduced serum levels of glucose (Active: 22.2 ± 2.9 mM, IBA: 12.8 ± 1.1 mM; Mann-Whitney test, $P < 0.001$) and insulin (Active: 44.4 ± 7.4 μU/mL, IBA: 8.4 ± 1.3 μU/mL; student's t-test, $P < 0.0001$; Fig. 1d, e and Extended Data Fig. 1). Consistent with the idea that fat becomes the primary energy source during hibernation, IBA squirrels exhibited increased levels of serum β-hydroxybutyrate (Active: 0.23 ± 0.04 mM, IBA: 2.07 ± 0.25 mM; student's t-test, $P < 0.0001$; Fig. 1f). Thus, despite many months of fasting and extensive utilization of internal fat resources, hibernating squirrels exhibit negligible food consumption (anorexia).

### Hibernating squirrels demonstrate reversible resistance to ghrelin

During fasting, the stomach releases the orexigenic hormone ghrelin, which activates Agouti-Related Peptide/Neuropeptide Y (AgRP/NPY) neurons in the arcuate nucleus of the hypothalamus (ARC), stimulating food consumption[12–16]. We tested whether anorexia during hibernation is caused by low levels of ghrelin. However, we found no significant difference in total and acylated (active) forms of ghrelin in blood plasma between IBA and active squirrels; instead, acylated ghrelin showed a trend towards increasing during hibernation (Fig. 2a–c). Because the high ghrelin levels observed during IBA are sufficient to induce feeding in active animals, we hypothesized that hibernating squirrels develop seasonal ghrelin resistance[8,17]. We tested this by monitoring food consumption in active and IBA squirrels after peripheral injection of ghrelin. As expected, ghrelin potentiated feeding in active squirrels to levels observed in mice and rats[18–20], and exceeding those of active squirrels after 48 h of food deprivation (Active PBS: 0.3 ± 0.2 g, Active Ghrelin: 2.3 ± 0.3 g; Two-Way ANOVA with Tukey's post hoc test, Active PBS v Active Ghrelin, $P < 0.001$; Fig. 2d, e). In stark contrast, ghrelin failed to potentiate food consumption when injected during IBAs, suggesting ghrelin resistance (IBA PBS: 0.2 ± 0.1 g, IBA Ghrelin: 0.4 ± 0.1 g; Two-Way ANOVA with Tukey's post hoc test, $P = 0.14$; Fig. 2d). To further challenge this conclusion, we compared the effect of ghrelin injection in IBA animals during hibernation and in the same animals after they became active after hibernation arousal. Active animals showed almost six-fold elevated food consumption compared to when they were in IBA, further strengthening the notion of ghrelin resistance during IBA (IBA ghrelin: 0.4 ± 0.1 g, Active ghrelin: 2.5 ± 0.3 g; paired student's t-test, $P < 0.02$; Fig. 2f).

Immunohistochemical analysis of cFOS expression showed that ghrelin injections activated a subset of ARC neurons in active, but not IBA, animals (Fig. 2g–i), suggesting that ARC neurons have reduced sensitivity to ghrelin during hibernation. In normal physiological conditions, ghrelin binds to growth-hormone secretagogue receptors (Ghsr) on AgRP/NPY neurons, triggering a release of the AgRP peptide from nerve terminals. Therefore, AgRP is predominantly found in nerve terminals rather than neuronal soma[13,21–25], a pattern we observed in active animals (Fig. 2j). In contrast, and in agreement with the idea of diminished sensitivity to ghrelin during IBA, AgRP accumulated in neuronal somas of IBA animals (Active: 0.8 ± 0.5, IBA: 47.4 ± 4.9 cell bodies, Mann-Whitney test, $P < 0.0001$, Fig. 2j, k and Extended Data Fig. 2), implying a diminished release of the peptide from these neurons.

### Hibernating squirrels have reduced leptin signaling

To investigate whether additional mechanisms contribute to hibernation anorexia, we asked if ARC neurons are sensitive to the satiety hormone leptin during IBAs. We found that plasma levels of leptin were slightly elevated in IBAs compared to active animals (Fig. 3a). However, during IBA ARC neurons showed reduced levels of the phosphorylated form of the signal transducer and activator of transcription 3 (pSTAT3) – a marker for leptin signaling in neurons[26] (Fig. 3b, c). Furthermore, we observed a decrease in the abundance of pSTAT3+ cells expressing cFOS (Fig. 3b–e). A subset of leptin-responsive neurons is marked by the expression of pro-opiomelanocortin (POMC) peptide and is responsible for producing satiety[27,28]. We observed a reduction in the abundance of POMC+ and POMC+/cFOS+ neurons during IBA, suggesting reduced activity in these cells (Fig. 3f–i). Together, these data show that leptin signaling in ARC neurons is reduced in hibernating squirrels.

### Ghsr and Lepr expression and BBB function are unaltered during IBA

To understand the mechanism for reduced ghrelin and leptin signaling during hibernation, we performed single cell sequencing of arcuate nucleus and median eminence (ARC-ME) neurons from active and IBA squirrels. 88,304 cells from ARC-ME were captured and analyzed after quality control (Active: 48,920 cells from 3 animals; IBA: 39,384 cells from 3 animals). ARC-ME tissue from squirrels in both states contained major cell types expected to be present in this brain area (see below). Further sub-clustering of neurons identified major neuronal populations, similar to those found in the ARC of mice[23], including *Pomc/Cartpt*, *Agrp/Npy*, *Kiss1*, *Ghrh*, *Sst*, and *Th*-expressing subclusters (Fig. 4a, c and Extended Data Table 1). AgRP and POMC neuronal clusters expressed known markers of these populations, confirming their identity[29] (Fig. 4b).

Based on our single-cell dataset, we found that the levels of ghrelin receptors in AgRP neurons and leptin receptors in POMC neurons remained unchanged between active and IBA animals (Fig. 4d, f). Furthermore, over 95% of *Ghsr* and over 89% of *Lepr* transcripts cloned de novo from the ARC represented functional isoforms[24,30] (Fig. 4e, g). Next, we tested whether blood brain barrier (BBB) function was impaired during hibernation, which would reduce peripheral hormone transport to the brain. Injections of 3 kDa dextran and 860 Da biocytin into active and IBA animals showed that ARC BBB retained its integrity during hibernation (Fig. 4h–j and Extended Data Fig. 3). Thus, the reduced ghrelin and leptin signaling in the ARC cannot be entirely attributed to a lack of functional receptors or impaired BBB, suggesting other mechanisms.

### Hibernating animals demonstrate central hypothyroidism

Our findings that ARC neurons resist the orexigenic effects of ghrelin and exhibit reduced leptin signaling during IBA suggested that activity of the hypothalamic feeding center is temporarily suppressed during hibernation. To understand the mechanism of this suppression, we turned our attention to the thyroid hormone triiodothyronine (T3), which stimulates food intake by acting on hypothalamic nuclei[31–35]. It has further been shown that the excitability of ARC AgRP neurons during fasting or ghrelin administration is increased by central thyroid hormone via uncoupling protein 2 (UCP2)-dependent mitochondrial proliferation[36,37]. We therefore hypothesized that anorexia during IBA may be due to central T3 deficiency.

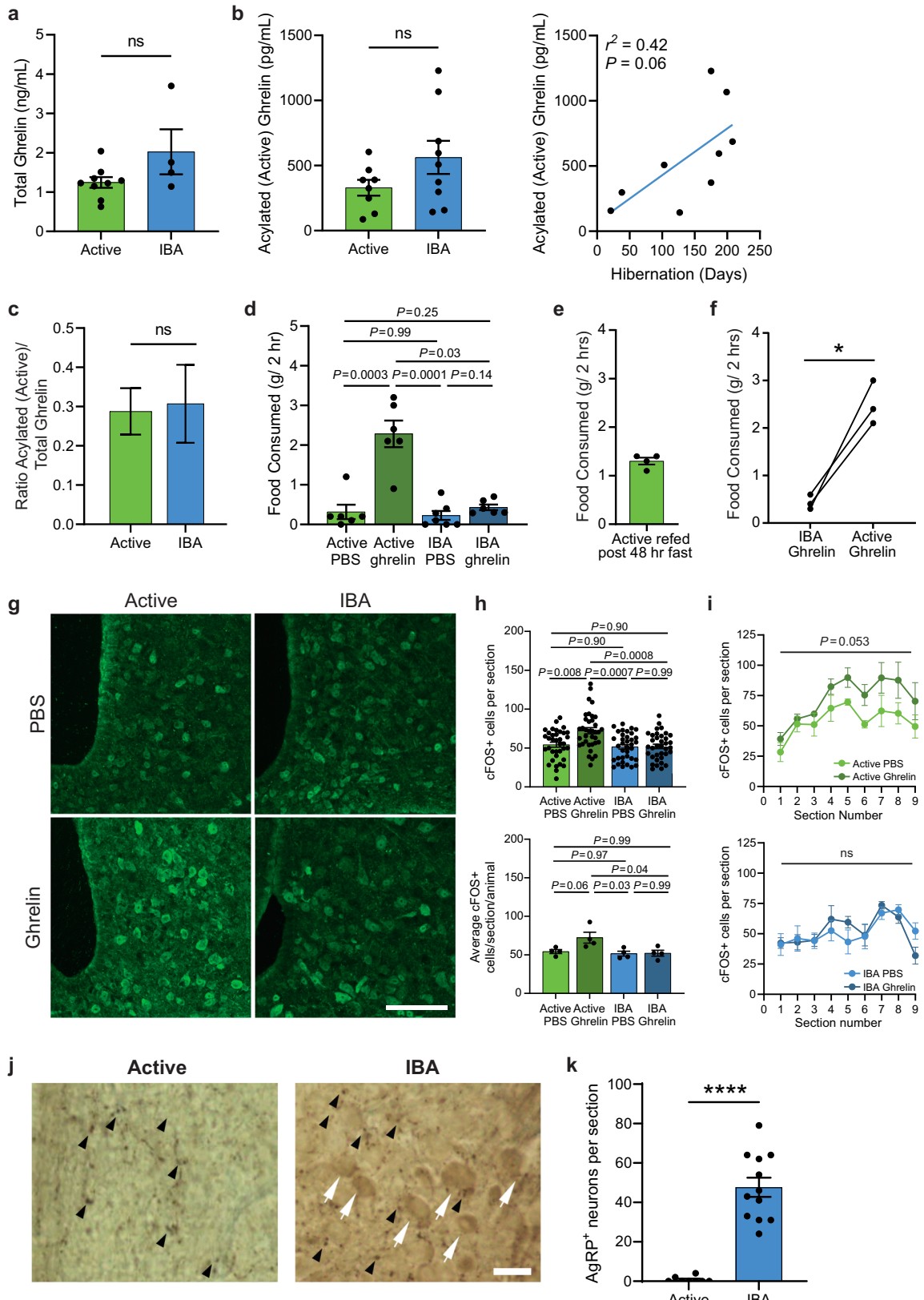

The translocation of T3 and its precursor T4 from the circulation to neurons occurs, in part, via the monocarboxylate transporter 8 (MCT8) expressed in endothelial cells and tanycytes[38–40]. Accordingly, our single cell RNA sequencing revealed MCT8 expression in these cell types in squirrel ARC (Fig. 5a, b). Furthermore, during IBA we observed a significant decrease in *MCT8* expression in tanycytes, but not in

neurons or endothelial cells (Fig. 5c–h). Next, we assessed the level of expression of the immunoglobulin superfamily member 1 (*Igsf1*). Loss of function mutations in IGSF1 causes central hypothyroidism in humans and mice[41–45]. *Igsf1* was expressed in neurons and astrocytes of squirrel ARC, and the level of expression significantly decreased in both groups during IBA (Extended Data Fig. 4). These results suggest a

**Fig. 2 | Hibernating squirrels demonstrate reversible resistance to ghrelin.**
Plasma levels of ghrelin across states. **a** Total ghrelin ($n = 9$ Active and $n = 4$ IBA
animals, mean ± SEM, Two-sided student's t-test, $P = 0.08$). **b** Acylated (active)
ghrelin across states (left, mean ± SEM, Two-sided student's t-test, $P = 0.13$) and the
same data plotted throughout hibernation (right, simple linear regression) ($n = 8$
Active and $n = 9$ IBA animals). **c** Ratio of acylated (active):total ghrelin (mean ± SEM,
Two-sided student's t-test, $P = 0.86$, calculated from **a** and **b**). **d** Two-hour food
consumption after ghrelin or PBS injection across states. $n = 6$ animals for Active
PBS, Active ghrelin, and IBA ghrelin and $n = 7$ animals for IBA PBS, mean ± SEM,
Two-Way ANOVA on rank-transformed data followed by Tukey's multiple com-
parisons, State main effect ($F_{1,21} = 5.82$, $P = 0.025$); Treatment main effect
($F_{1,21} = 26.82$, $P < 0.0001$); Interaction ($F_{1,21} = 4.10$, $P = 0.056$). **e** Two-hour food
consumption of Active animals following a 48-h fast ($n = 4$ animals, mean ± SEM).
**f** Two-hour food consumption of animals during hibernation and following arousal
into the active season ($n = 3$ animals, Two-sided student's paired t-test, $P = 0.02$).
**g** Representative images of ARC cFOS staining of active and IBA squirrels injected
with ghrelin or PBS control. Arrowheads, cFOS+ cells. Scale bar, 100 μm.
**h** Quantification of cFOS+ cells per section across ARC volume (top) and average
cFOS+ cells per section per animal (bottom) across states after ghrelin injection
($n = 36$ sections per group from 9 serial sections per 4 independent biological
replicates, mean ± SEM, (top) Two-Way ANOVA on rank transformed data followed

by Tukey's multiple comparisons, State main effect ($F_{1,140} = 5.25$, $P = 0.024$);
Treatment main effect ($F_{1,140} = 10.73$, $P = 0.001$); Interaction ($F_{1,140} = 5.23$,
$P = 0.024$), (bottom) Two-Way ANOVA followed by Tukey's multiple comparisons,
State main effect ($F_{1,12} = 6.44$, $P = 0.026$); Treatment main effect ($F_{1,12} = 4.24$,
$P = 0.062$); Interaction ($F_{1,12} = 3.73$, $P = 0.077$)). **i** cFOS+ cells per section throughout
9 serial sections taken across the volume of the ARC for Active (top) and IBA
(bottom) animals after control or ghrelin treatment ($n = 4$ independent animals per
group). Data are the same as **h** (top) replotted by section number (mean ± SEM),
Two-Way ANOVA followed by Tukey's multiple comparisons, (top) Section number
main effect ($F_{2.4,14.4} = 6.67$, $P = 0.007$); Treatment main effect ($F_{1,6} = 5.75$, $P = 0.053$);
Interaction ($F_{8,48} = 0.54$, $P = 0.82$), (bottom) Section number main effect
($F_{3.5,20.7} = 4.74$, $P = 0.009$); Treatment main effect ($F_{1,6} = 0.01$, $P = 0.91$); Interaction
($F_{8,48} = 1.12$, $P = 0.37$). **j** Representative immuno-EM images of ARC from active and
IBA animals. White arrows, neuronal soma; black arrowheads, fibers stained for
AgRP. Scale bar, 20 μm. **k** Quantification of AgRP+ neuronal soma per section.
($n = 8$ sections total from 3 Active animals and $n = 12$ sections total from 2 IBA
animals; mean ± SEM, Two-sided Mann-Whitney test, $P < 0.0001$). (**a–f, h** bottom)
Each point represents one animal. (**h** top, **k**) Each point represents one section. ns:
not significant $P > 0.05$; *$P < 0.05$; ****$P < 0.0001$. Source data are provided as a
Source Data file.

---

potential decrease in hypothalamic T3 during hibernation. In support
of this, direct measurements showed more than two-fold lower levels
of hypothalamic T3 during IBA compared to active animals (Active:
$0.77 ± 0.1$ pg/mg tissue, IBA: $0.32 ± 0.05$ pg/mg tissue; student's t-test,
$P < 0.01$; Fig. 6a). We also detected a steady decline in hypothalamic T3
during the active season, reaching the level observed during IBA by the
end of hibernation (Fig. 6a). At the same time, the level of T3 in blood
serum remained unchanged, demonstrating that thyroid hormone
deficiency during hibernation was restricted to the CNS (Fig. 6b). We
also found significantly higher blood serum levels of T4 during IBA
(Fig. 6c), further supporting the idea that hibernating squirrels exhibit
central, but not peripheral, hypothyroidism.

Thyroid hormones canonically exert their action by binding to
nuclear thyroid hormone receptors alpha (*Thra*) and beta (*Thrb*)[46].
Single-cell sequencing of ARC neurons revealed that *Thra* and *Thrb* are
expressed in AgRP and POMC neurons, and that their expression levels
are similar in both physiological states (Fig. 6d, e), suggesting that
hypothyroidism in IBA animals is not caused by receptor
downregulation.

### Central T3 infusion rescues hibernation anorexia

Our data suggest that anorexia during hibernation could be caused by
hypothalamic T3 deficiency. To test this hypothesis, we bypassed this
transport step by infusing T3 directly into the mediobasal hypothala-
mus during IBA and measured its effect on feeding (Fig. 6f). Remark-
ably, while T3 injection did not induce feeding during the first two
hours post-injection (Fig. 6g, h, left), it resulted in robust and sig-
nificant potentiation of feeding over a 24-h period (Hibernation
Season 1: Control: $1.6 ± 0.3$ g, 15.3 nmol T3: $2.5 ± 0.3$ g; paired student's t-test,
$P < 0.001$; Hibernation Season 2: Control: $1.6 ± 0.6$ g, 15.3 pmol T3:
$3.3 ± 0.9$ g; paired student's t-test, $P = 0.02$; Fig. 6g, h, right), consistent
with its role as a transcriptional regulator. Thus, these data strongly
suggest that the specific deficiency of T3 in the CNS contributes to
reversible anorexia during hibernation.

## Discussion

Hibernation was first documented in 350 BCE by Aristotle, who noted
that some creatures cease eating and conceal themselves in a sleep-like
state for many months to pass the winter[47]. Hibernation invokes a series
of flexible adaptations that allow animals to thrive in inhospitable
environments, where they experience thermal challenges and food
scarcity[48]. Coordination of hunger and satiety is essential for hibernators
to survive, as premature emergence from underground burrows to seek

food may dysregulate dependent processes and increase the risk of
predation. In this study, we found that squirrels exhibit T3 deficiency in
the hypothalamus during IBA and that restoration of T3 in the hypo-
thalamus reverses anorexia. One limitation of our study is that a direct
infusion of T3 into the hypothalamus creates a high local concentration
of the hormone. Nevertheless, that supraphysiological T3 infusion is
capable of opposing hibernation-induced anorexia, strongly supports
the idea that long-term suppression of hunger during hibernation is, at
least partially, due to central hypothyroidism. We showed, using single
cell sequencing, that the amount of *Ghsr* transcript does not decrease
during hibernation. However, there remains a possibility that ghrelin
receptor protein is functionally downregulated and thus may also con-
tribute to seasonal anorexia.

Our findings demonstrate that thyroid hormone deficiency during
hibernation is restricted to the CNS and does not extend to peripheral
levels of T3 and T4. Hibernating squirrels thus present a remarkable
animal model in which central thyroid hormone function is depressed
while essential peripheral thyroid function on internal organs and
general metabolic processes is preserved[49,50]. Central thyroid hormone
has been implicated in the seasonal shifts in reproduction and food
intake in non-hibernating animals, including Siberian hamsters, sheep,
photoperiodic F344 rats[51–54] and hibernating arctic ground squirrels[55].

Although the precise mechanism of central hypothyroidism
remains to be determined in our model, the data presented here
suggest a putative mechanism with two complimentary steps. The first
step involves a reduction in MCT8 expression, leading to limited
transport of thyroid hormone from circulation into the hypothalamus.
This finding echoes data in humans and mice, where loss of functional
MCT8 leads to central hypothyroidism, causing a panel of metabolic
and neurological abnormalities[56–58]. The second step involves IGSF1, a
protein expressed in squirrel hypothalamic neurons and astrocytes.
IGSF1 dysfunction is strongly linked with congenital central hypo-
thyroidism in humans, and this phenotype is recapitulated in mouse
models[41–45]. Although the exact mechanism of IGSF1 is a matter of
intense research, our data suggest that reversible IGSF1 deficiency in
hypothalamic neurons could be part of the natural seasonal physiology
of squirrels, contributing to reversible anorexia during hibernation.
Our data show that, in contrast to humans, central hypothyroidism in
squirrels is an essential component of normal physiology. In both
cases, however, this process relies on similar molecular pathways,
suggesting hibernating squirrels as a naturalistic model to study the
mechanism of central hypothyroidism and associated diseases in
humans.

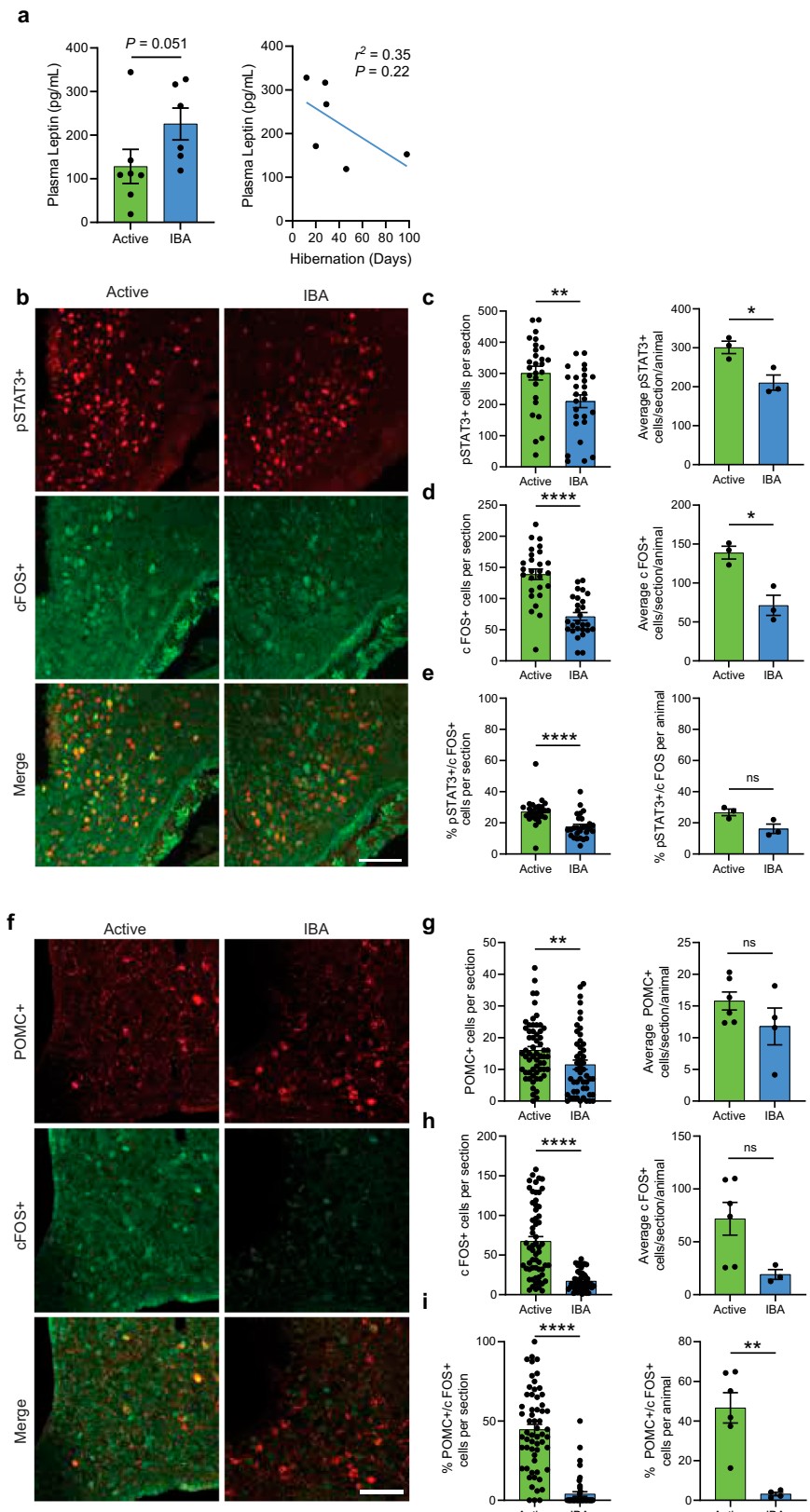

## Methods

### Animals

All experimental procedures were performed in compliance with the Institutional Animal Care and Use Committee of Yale University (protocol 2021-11497). Thirteen-lined ground squirrels (*Ictidomys tri-decemlineatus*) were obtained from Dr. Dana Merriman (University of Wisconsin-Oshkosh) and/or bred in our facilities (Yale University). Animals (age 0.5–3 years) of both sexes were single housed in

**Fig. 3 | Hibernating squirrels have reduced leptin signaling. a** Plasma leptin levels across states (left, mean ± SEM, Two-sided Mann-Whitney test, $P = 0.051$) and throughout hibernation (right, simple linear regression) ($n = 7$ active and $n = 6$ IBA animals). **b** Representative immunohistochemistry images of ARC in active and IBA squirrels using anti-pSTAT3 and anti-cFOS antibody. **c–e** Quantification of **c** pSTAT3+ cells, **d** cFOS+ cells, and **e** percent pSTAT3+ cells colocalizing cFOS by state. (Left) Number of positive cells per section (mean ± SEM, **c** Two-sided student's $t$-test, $P = 0.004$, **d** Two-sided student's $t$-test, $P = 3.2 \times 10^{-8}$, **e** Two-sided Mann-Whitney test, $P = 1.58 \times 10^{-7}$. Each point is one section). (Right) Average number of positive cells per section per animal (mean ± SEM, Two-sided student's $t$-test, **c** $P = 0.02$, **d** $P = 0.01$, **e** $P = 0.07$. Each point is one animal). ($n = 27$ sections total from 3 Active animals and $n = 27$ sections total from 3 IBA animals). **f** Representative immunohistochemistry images of ARC in Active and IBA squirrels using anti-POMC and anti-cFOS antibody. **g–i** Quantification of **g** POMC+ cells, **h** cFOS+ cells, and **i** percent POMC+ cells colocalizing with cFOS by state. (Left) Number of positive cells per section (mean ± SEM, Two-sided Mann-Whitney test, **g** $P = 0.004$, **h** $P < 0.0001$, **i** $P < 0.0001$. Each point is one section). (Right) Average number of positive cells per section per animal (mean ± SEM, Two-sided student's $t$-test, **g** $P = 0.20$, **h** $P = 0.06$, **i** $P = 0.002$. Each point is one animal). ($n = 59$ sections total from 6 Active animals and $n = 50$ sections total from 4 IBA animals). ns: not significant $P > 0.05$; *$P < 0.05$; **$P < 0.01$; ****$P < 0.0001$. Scale bars, 100 μm. Source data are provided as a Source Data file.

temperature- and humidity-controlled facilities at Yale University. All squirrels were implanted with an interscapular temperature transponder (IPTT-300, BMDS).

During the active season (May–August), squirrels were kept in a vivarium at $21 \pm 1\,°C$ under a 12 h:12 h light:dark cycle at 40–60% humidity and maintained on a diet of dog food (IAMS) supplemented with sunflower seeds, superworms, and fresh vegetables (celery and carrots) with *ad libitum* access to water.

During the fall pre-hibernation season (September–October), squirrels were kept in a vivarium at $20 \pm 1\,°C$ under a light:dark cycle matched to Central Standard Time sunrise:sunset, which corresponds to the native time zone of thirteen-lined ground squirrels. Animals were kept at 40–60% humidity and maintained on a diet of dog food (IAMS) supplemented with sunflower seeds with *ad libitum* access to water.

During hibernation season (September–April), hypothermic squirrels (body temperature ~$20 \pm 1\,°C$) were moved to the hibernaculum, which was kept at $4\,°C$ at 40–60% humidity under constant darkness, without access to food or water.

In this study, "active" squirrels were those with a constant core body temperature (CBT) of $37\,°C$ during the active season. "Pre-hibernation" squirrels were those that generally maintained euthermia, but also demonstrated transient, hypothermic bouts to ~$20\,°C$. "IBA" squirrels were those who had undergone at least one bout of hypothermic torpor during the hibernation season but had achieved a CBT of > $32\,°C$ for ≥ 60 min, or ≥ 20 min for the central thyroid hormone experiments.

### Food consumption and body mass measurement
Food consumption and body weight of adult animals were measured every 2 weeks during the active period (May-August) and into the pre-hibernation season (September–October). Active and pre-hibernation animals were moved to the behavioral room kept at $20 \pm 1\,°C$ under a 12 h:12 h light:dark cycle and acclimated overnight. In the morning (9–11 AM), each squirrel was weighed, transferred to a clean cage, and allowed to habituate for 30 min. Food consumption measurements were performed with dog food only. Pre-weighed food was added to each cage, and animals were allowed to feed undisturbed. Food remaining 24 h later was weighed and used to calculate daily food consumption. A separate bowl of dog food was kept in the behavior room to control for food weight changes due to ambient humidity, but no difference was found so no correction was needed. The maximum food consumption per animal for the active season was reported. Pre-hibernation feeding data corresponds to the last 24-h period in the pre-hibernation season where animals maintain euthermia for the duration of the experiment.

Hibernating animals entered IBA spontaneously, so their food consumption and body weight measurements occurred between 11 AM and 8 PM. IBA animals were weighed, transferred to a clean cage in the hibernaculum kept at $4\,°C$ under constant darkness and allowed to habituate for 30 min. Food consumption measurements were performed with dog food only. Pre-weighed food was added to each cage, and animals were allowed to feed undisturbed. Food remaining 24 h later was weighed and used to calculate daily food consumption. The daily food consumption of animals whose body temperature dropped below $32\,°C$ within the 24-h testing window were excluded from analysis. The body weight of excluded animals was included. IBA measurements occurred just once during the hibernation season, to ensure that animals remained naïve to food availability during the winter. A separate bowl of dog food was kept in the behavior room, kept at $4\,°C$ in constant darkness, to control for weight changes due to ambient humidity, but no difference was found so no correction was needed.

### Blood collection
Animals were euthanized by isoflurane overdose. The chest cavity was opened, the right atrium of the heart pierced, and trunk blood was collected with a 18 G needle and syringe.

### Serum hormone and metabolite measurements
Whole blood was allowed to coagulate at room temperature for 30 min, then centrifuged at $4\,°C$ at 2000 x g for 15 min. Serum was aliquoted and stored at $-80\,°C$ for later use. Serum glucose, insulin, beta-hydroxybutyrate, total T3, and total T4 measurement were performed by Antech Diagnostics (Fountain Valley, CA).

### Plasma ghrelin measurements
Whole blood was collected into chilled, pre-coated $K_3$ EDTA tubes (MiniCollect, Grenier Bio-One) and immediately treated with Pefabloc (Sigma) to a final concentration of 1 mM. Blood was centrifuged at 1600 x g for 15 min. Plasma was aliquoted and stored at $-80\,°C$ for later use. Plasma active (acylated) ghrelin was measured by mouse/rat ELISA (EZRGRA-90K, Millipore). Plasma total ghrelin was measured by mouse/rat ELISA (EZRGRT-91K, Millipore). All samples were run in duplicate. The ratio of acylated (active form of ghrelin)/ total ghrelin was calculated by dividing the mean of the acylated ghrelin concentration by the mean of the total ghrelin concentration per state. The SEM of the ratio was calculated by simple error propagation given by the formula:

$$\sigma_{ratio} = A/B * sqrt((\sigma_A/A)^2 + (\sigma_B/B)^2)$$

where A and B are mean values of active (acylated) and total ghrelin, respectively.

### Plasma leptin measurements
Whole blood was collected into chilled, pre-coated $K_2$EDTA tubes (BD Vacutainer, Lavender/H) and immediately treated with aprotinin (Millipore Sigma, 9087-70-1) to a final concentration of 0.02 mM. Blood was centrifuged at 1600 x g for 15 min. Plasma was aliquoted and stored at $-80\,°C$ for later use. Plasma leptin was measured by mouse/rat ELISA (R&D Systems, MOB00). All samples were run in duplicate. A ROUT outlier test ($Q = 1\%$) was run to identify one outlier in the Active state and two outliers in the IBA state.

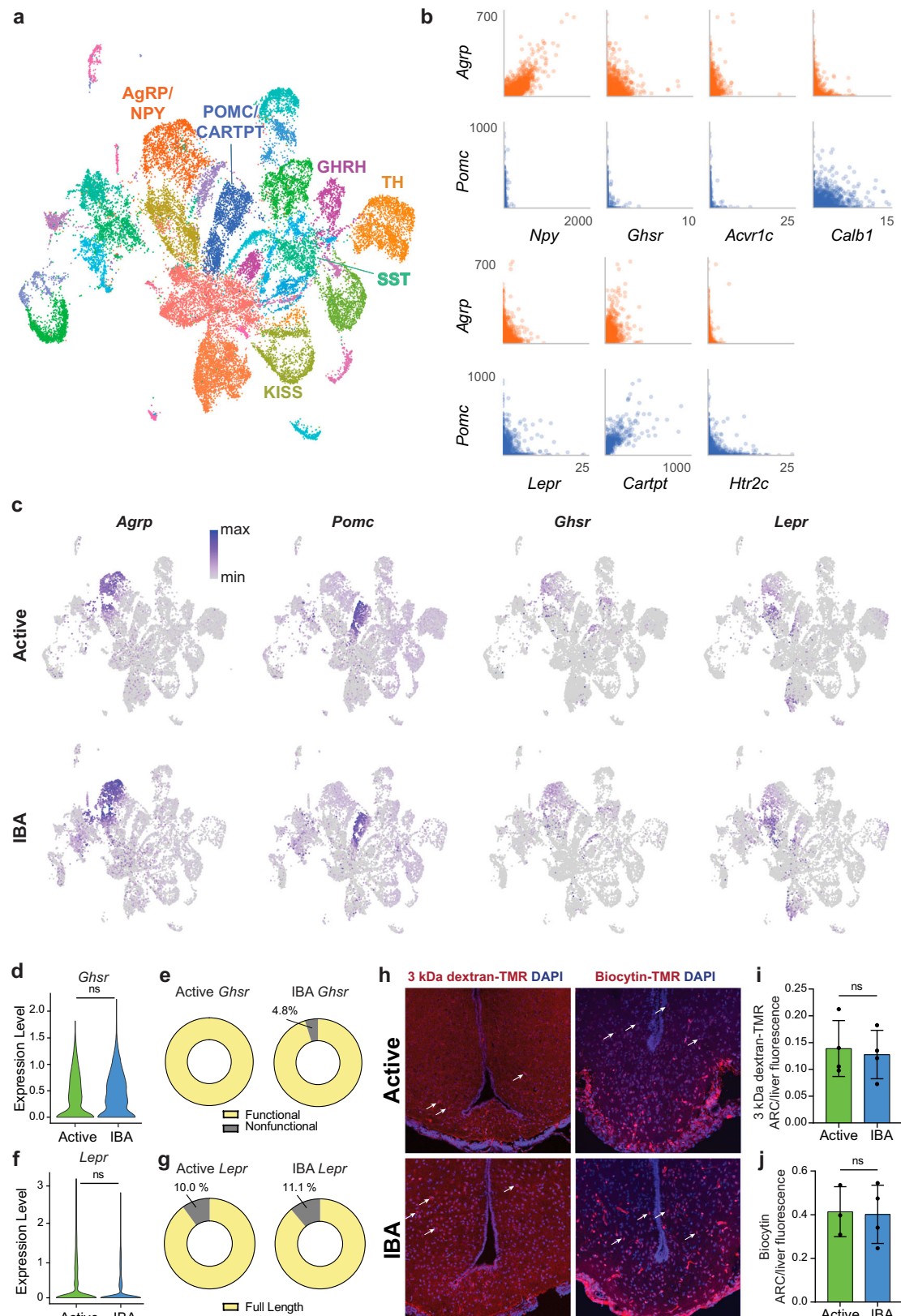

## Intraperitoneal Ghrelin injections

Ground squirrels were acclimated in the behavioral room overnight. Animals were weighed, transferred to clean cages, and allowed to habituate for 30 min. Squirrels were immobilized with decapicones, injected with 2 mg/kg acylated rat ghrelin (1465, Tocris) solubilized in

PBS using an injection volume of 2 mL/kg body weight, and returned to their cages. Control injections were PBS injected at a volume of 2 mL/kg body weight. Pre-weighed food was added to the cage and animals allowed to feed for two hours. The food remaining after the feeding period was weighed and used to calculate food consumption.

**Fig. 4 | Ghsr and Lepr expression and BBB function are unaltered during IBA.**
**a** 2-dimensional UMAP projection of gene expression in individual arcuate nucleus and median eminence (ARC-ME) neurons colored by cluster (*n* = integration from 3 Active and *n* = integration from 3 IBA animals). **b** Coexpression of *Agrp* and *Pomc* with known markers of AgRP and POMC neurons. Values are in counts per 10,000 total counts. **c** Expression of *Agrp, Pomc, Ghsr,* and *Lepr* genes in individual ARC-ME neurons in Active and IBA states (normalized, log-transformed, and represented by color as indicated in the color bar). **d** Expression of *Ghsr* in AgRP neurons across states (Two-sided Wilcoxon rank sum test (R/Seurat), *P* > 0.05). **e** Quantification of functional versus nonfunctional *Ghsr* isoforms from de novo cloning across states (*n* = 38 clones total from 2 Active animals and *n* = 42 clones total from 2 IBA animals). **f** Expression of *Lepr* in POMC neurons across states (Two-sided Wilcoxon

rank sum test (R/Seurat), *P* > 0.05). **g** Quantification of truncated versus full length long-form leptin-receptor from de novo cloning across states (*n* = 10 clones total from 2 Active animals and *n* = 9 clones total from 2 IBA animals). **h–j** Blood brain permeability assays by tail artery injection in Active and IBA squirrels of 3 kDa dextran-tetramethylrhodamine (TMR) and 860 Da biocytin-TMR. **h** Representative images of ARC-ME demonstrating deposition of dye (white arrows). Scale bar, 50 μm. **i** Quantification of normalized fluorescence of 3 kDa dextran-TMR (*n* = 4 Active and *n* = 4 IBA animals, mean ± SEM, Two-sided student's *t*-test, *P* = 0.76) and **j** 860 Da biocytin-TMR (*n* = 3 Active and *n* = 4 IBA animals, mean ± SEM, Two-sided student's *t*-test, *P* = 0.90). Each point represents one animal. ns: not significant *P* > 0.05. Source data are provided as a Source Data file.

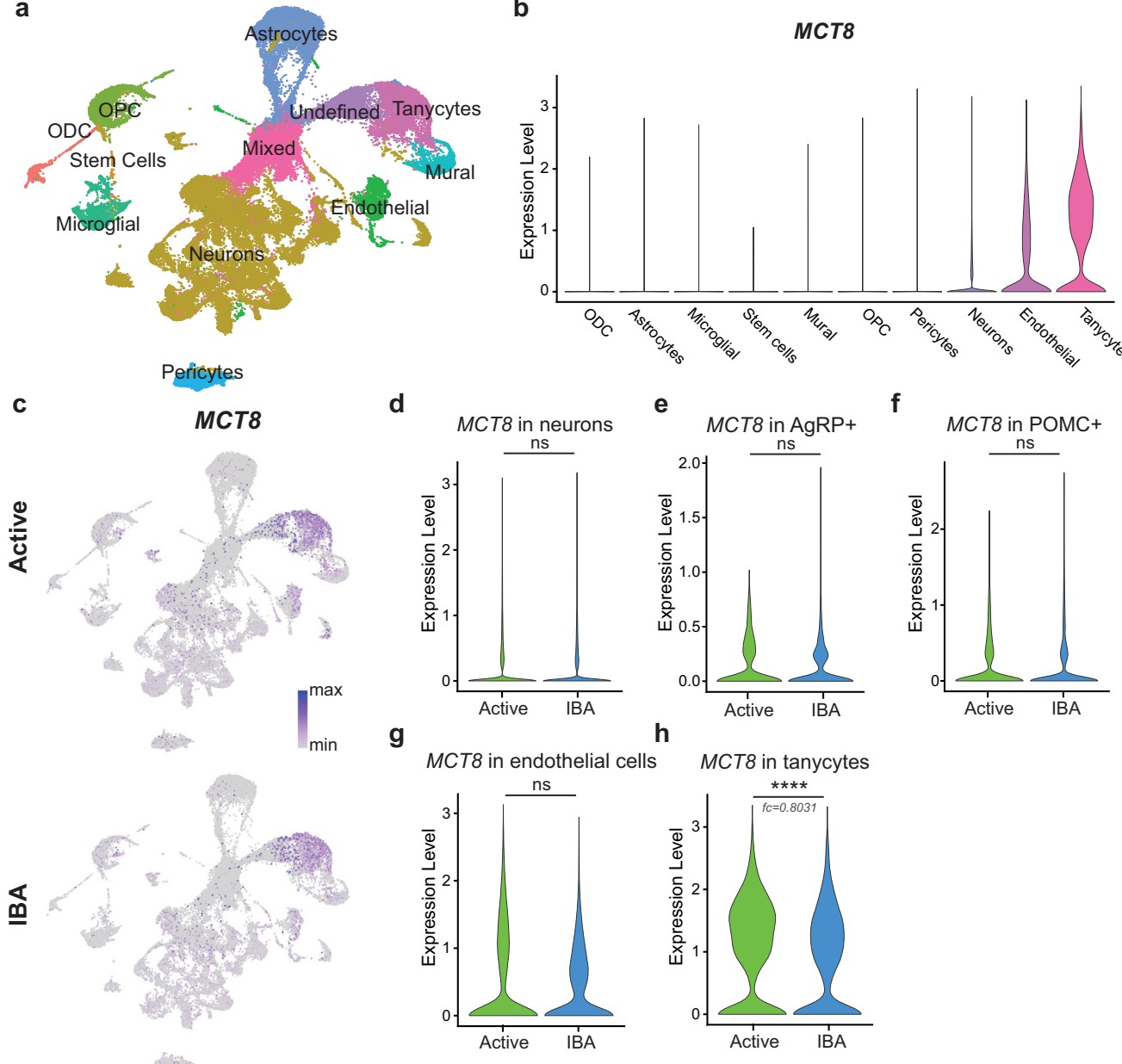

**Fig. 5 | Thyroid hormone transporter *MCT8* is downregulated in tanycytes during hibernation. a** 2-dimensional UMAP projection of gene expression in individual arcuate nucleus and median eminence (ARC-ME) cells, colored by cell type, as determined by clustering and annotation. **b, c** Expression of *MCT8* in identified cell types, aggregated for both states (**b** violin plots) and separated by

state (**c** UMAP plots). **d–h** Expression of *MCT8* across states in **d** all neurons, **e** AgRP neurons, **f** POMC neurons, **g** endothelial cells, and **h** tanycytes. Violin plots show normalized log-transformed gene counts. (Two-sided Wilcoxon rank sum test (R/Seurat), **d–g** *P* > 0.05, **h** *P* = 6.8 × 10⁻¹⁷). ns: not significant *P* > 0.05; ****P* < 0.0001; fc: fold change.

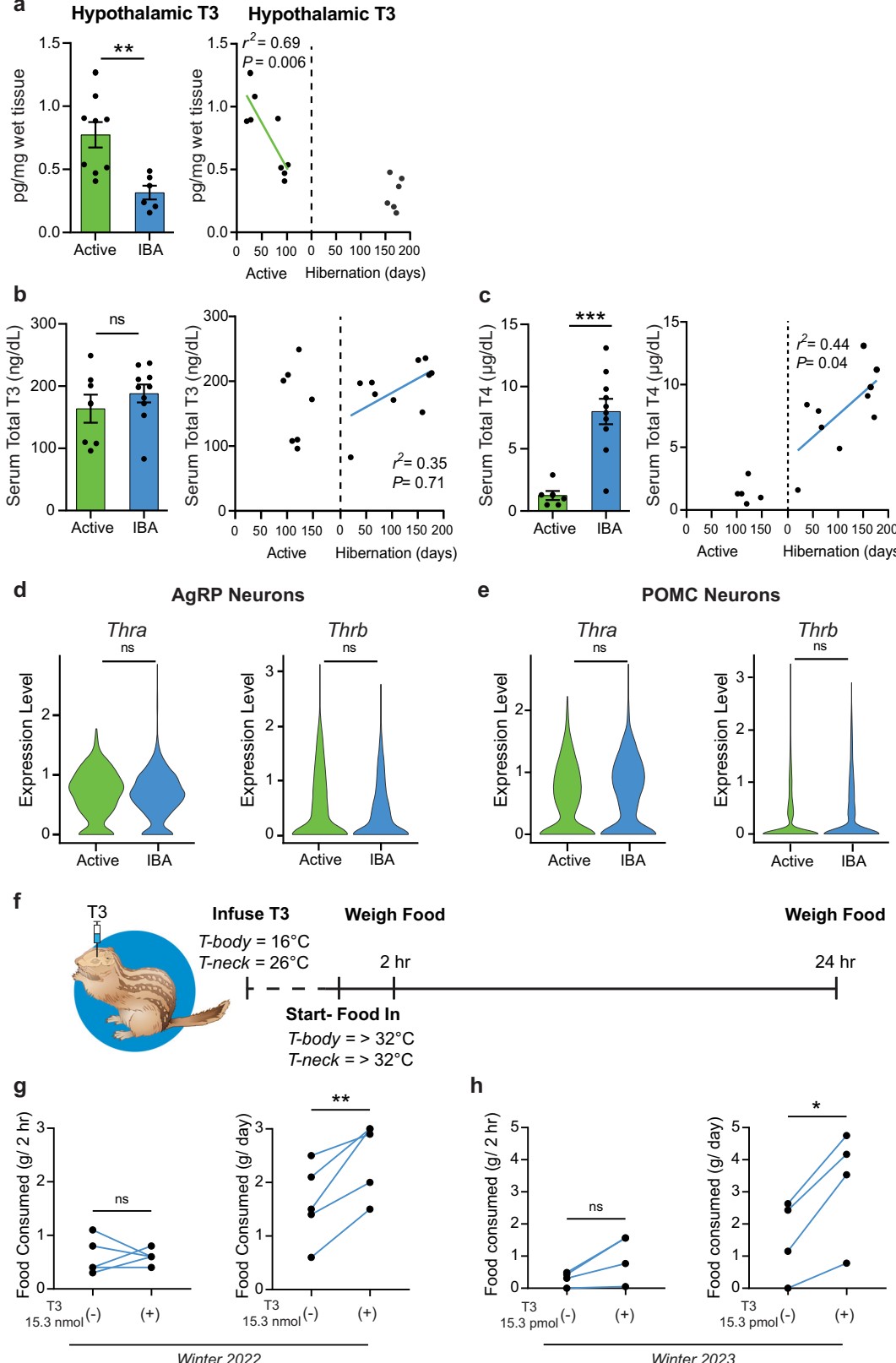

## Immunohistochemistry

Ground squirrels were deeply anesthetized by isoflurane inhalation and then subjected to intracardiac perfusion with PBS followed by fixative (4% paraformaldehyde in PBS). Brains were post-fixed overnight, and transferred to serial 10%, 20%, 30% sucrose solutions after sinking. Brains were embedded in OCT, frozen on dry ice, and stored at

−80 °C until use. Coronal brain sections of the arcuate nucleus were cut at a thickness of 40 µm on a cryostat (Leica, CM3050S). Sections were mounted onto SuperFrost Plus slides and stored at −80 °C with desiccant until the day of the immunohistochemistry procedure. Sections were dried in an incubator at 37 °C for 30 min. Slides were washed three times with PBS for 10 min, and then washed with 1%

**Fig. 6 | Reversible central hypothyroidism underlies hibernation anorexia. a** T3 content measured from homogenized hypothalamus from Active and IBA squirrels (left, mean ± SEM, Two-sided student's *t*-test, *P* = 0.004), and plotted by days active and days in hibernation (right, simple linear regression) (*n* = 9 Active and *n* = 6 IBA animals). Serum total thyroid hormone concentration across active and IBA seasons (left, mean ± SEM) and plotted by days active and days in hibernation (right, simple linear regression). **b** Total T3 (*n* = 7 Active and *n* = 10 IBA animals, Two-sided student's t-test, *P* = 0.35). **c** Total T4 (*n* = 6 Active and *n* = 10 IBA animals, Two-sided student's *t*-test, *P* = 0.0002). **d**–**e** Expression of *Thra* and *Thrb* in **d** AgRP neurons and **e** POMC neurons by single cell RNA sequencing. Violin plots show normalized log-transformed gene counts. Two-sided Wilcoxon rank sum test (R/Seurat,

*P* > 0.05). **f**–**h** Hypothalamic infusion of T3 during IBA and resulting food consumption. **f** Schematic of hypothalamic T3 infusion during IBA after arousal from torpor. The squirrel image is adapted from the authors' previous publication[48] and the brain schematic is available under the Creative Commons CC0 1.0 Universal Public Domain Dedication. Paired food consumption at two-hours (left) and 24-hours (right) after control and hypothalamic infusion of **g** 15.3 nmol T3 (*n* = 5 animals, Two-sided paired student's t-test, (left) *P* > 0.99, (right) *P* = 0.01) and **h** 15.3 pmol T3 (*n* = 4 animals, Two-sided paired student's *t*-test, (left) *P* = 0.08, (right) *P* = 0.02). Each point represents one animal. ns: not significant *P* > 0.05; *\*P* < 0.05; *\*\*P* ≤ 0.01; *\*\*\*P* < 0.001. Source data are provided as a Source Data file.

H2O2 and 1% NaOH in PBS for 10 min. Slides were moved to 0.3% glycine in PBS 1x for 10 min and washed with 0.03% SDS in PBS. Sections were blocked for two hours at room temperature with 5% normal goat serum in 0.5% PBST.

For cFOS immunohistochemistry sections were incubated with primary antibody (1:500, mouse monoclonal cFOS C-10, Santa Cruz, sc-271243) at 4 °C for 24 h. After incubation with the primary antibody, sections were washed four times with 0.1% PBST for 15 min. Sections were incubated with secondary antibody (1:400, Alexa Fluor 488 goat anti-mouse, Invitrogen, a11001) for two hours at room temperature. Sections were washed four times with 0.1% PBST for 15 min, followed by a wash with PBS.

For pSTAT3 immunohistochemistry sections were incubated with primary antibody (1:200, rabbit polyclonal Phospho-Stat3 (Tyr705), Cell Signaling Technology, 9131) at 4 °C for 24 h. After incubation with the primary antibody, sections were washed five times with 0.1 % PBST for 10 min. Sections were incubated with secondary antibody (1:1000, Alexa Fluor 555 goat anti-rabbit, Abcam, ab150086) for two hours at room temperature. Sections were washed five times with 0.1% PBST for 15 min, followed by a wash with PBS.

For POMC immunohistochemistry, sections were incubated with primary antibody (1:2000, porcine anti-rabbit polyclonal POMC, Phoenix Pharmaceuticals, H-029-030) at 4 °C for 24 h. After incubation with the primary antibody, sections were washed five times with 0.1% PBST for 10 min. Sections were incubated with secondary antibody (1:1000, Alexa Fluor 555 goat anti-rabbit, Abcam, ab150086) for two hours at room temperature. Sections were washed five times with 0.1% PBST for 15 min, followed by a wash with PBS.

Slides were mounted using Vectashield with DAPI. Sections were imaged on a Leica SP8 Confocal Microscope at 20X using LASX software. Negative controls (secondary antibody only) were performed for cFOS, pSTAT3, and POMC immunohistochemistry and showed no non-specific fluorescent binding.

**Immuno-electron microscopy**
Ground squirrels were deeply anesthetized by isoflurane inhalation and were subjected to intracardiac perfusion. Free-floating sections (50 μm thick) were incubated with rabbit anti-AgRP antibody (Phoenix Pharmaceuticals) diluted 1:2000 in 0.1 M PB after 1 h blocking in 0.1 M PB with 5% normal goat serum. After several washes with PB, sections were incubated in the secondary antibody (biotinylated goat anti-rabbit IgG; 1:250 in PB; Vector Laboratories Inc.) for two hours at room temperature, then rinsed several times in PB followed by incubation for two hours at room temperature with avidin–biotin–peroxidase (ABC; 1:250 in PB; VECTASTAIN Elite ABC kit PK6100, Vector Laboratories). The immunoreaction was visualized with 3,3-diaminobenzidine (DAB). Sections were then osmicated (1% osmium tetroxide) for 30 min, dehydrated through increasing ethanol concentrations (using 1% uranyl acetate in 70% ethanol for 30 min), and flat-embedded in Durcupan between liquid release-coated slides (product no. 70880, Electron Microscopy Sciences). After embedding in Durcupan (14040, Electron Microscopy Sciences), ultrathin sections were cut on a Leica Ultra-Microtome, collected on Formvar-coated single-slot grids, and

analyzed with a Tecnai 12 Biotwin electron microscope (FEI) with an AMT XR-16 camera.

**Primary cell dissociation and single-cell RNA sequencing**
Primary neurons were isolated from the arcuate nucleus of hypothalamus and median eminence following a published protocol[1] with modification. Animals were euthanized by isoflurane inhalation over-dose followed by cardiac perfusion with brain perfusion solution (containing in mM: 196 sucrose, 2.5 KCl, 28 NaHCO3, 1.25 NaH2PO4, 7 Glucose, 1 Sodium Ascorbate, 0.5 CaCl2, 7 MgCl2, 3 Sodium Pyruvate, oxygenated with 95% O2/5% CO2, osmolarity adjusted to 300 mOsm with sucrose, pH adjusted to 7.4). The brain was dissected and slices were cut on a vibratome (Leica, VT1200). A brain slice containing the ARC and ME were identified by the presence of the third ventricle and separation of the optic chiasm. Three successive 600-μm slices containing the ARC were collected. The area around the third ventricle was microdissected from the brain slices using a micro-scalpel (Fine Science Tools, 10055-12).

Tissue was digested in Hibernate A medium (custom formulation with 50 mM glucose and osmolarity adjusted to 280 mOsm, BrainBits) supplemented with 1 mM lactic acid (Sigma, L1750), 0.5 mM GlutaMAX (ThermoFisher, 35050061) and 2% B27 minus insulin (ThermoFisher, A1895601) containing 20 U/ml papain (Worthington Biochemical Corporation, LS003124) in a shaking water bath at 34 °C for 30 min and dissociated by mechanical trituration through the tips of glass Pasteur pipettes with decreasing diameter (0.9 mm, 0.7 mm, 0.5 mm, 0.3 mm). Cell suspension was centrifuged over 8% bovine serum albumin (Sigma, A9418-5G) layer. Supernatant was removed leaving ~50 μl of suspension. Cell suspension was resuspended in 950 μl of Hibernate A medium (same formulation as above) and centrifuged at 300 x g for 5 min. Supernatant was removed, leaving ~50 μl of cell suspension, which was gently mixed with a glass pipette and stored on ice. A 10 μl aliquot of cell suspension was mixed with 10 μl of Trypan Blue stain, loaded into a hemocytometer, and used to assess cell concentration and viability.

Cell suspension was processed according to the 10X Genomics library preparation protocol at the Center for Genome Analysis/Keck Biotechnology Resource Laboratory at Yale University. Single cell suspension in RT Master Mix was loaded on the Single Cell G Chip and partitioned with a pool of about 750,000 barcoded gel beads to form nanoliter-scale Gel Beads-In-Emulsions (GEMs). The volume of cell suspension for loading was calculated based on cell concentration to capture 10,000 cells. Upon dissolution of the Gel Beads in a GEM, the primers were released and mixed with cell lysate and Master Mix. Incubation of the GEMs produced barcoded, full-length cDNA from poly-adenylated mRNA. Silane magnetic beads were used to remove leftover biochemical reagents and primers from the post GEM reaction mixture. Full-length, barcoded cDNA was amplified by PCR to generate sufficient mass for library construction. Enzymatic Fragmentation and Size Selection were used to optimize the cDNA amplicon size prior to library construction. R1 (read 1 primer sequence) was added to the molecules during GEM incubation. P5, P7, a sample index, and R2 (read 2 primer sequence)

were added during library construction via End Repair, A-tailing, Adapter Ligation, and PCR. The final libraries contained the P5 and P7 primers used in Illumina bridge amplification. Sequencing libraries were sequenced on an Illumina NovaSeq instrument with 150 bp reads according to the manufacturer's instructions at the depth of ~1.1-1.4 billion reads/sample.

Raw sequencing reads were processed using 10X CellRanger v.6.1.2 (10X Genomics, Pleasanton, CA). Custom genome reference for thirteen-lined ground squirrel (*Ictidomys tridecemlineatus*) was built based on the reference genome sequence and annotation obtained from the Ensembl project (www.ensembl.org[2] Release 101; all files accessed on 11/20/2020):

Genome:

ftp://ftp.ensembl.org/pub/release-101/fasta/ictidomys_tridecemlineatus/dna/Ictidomys_tridecemlineatus.SpeTri2.0.dna.toplevel.fa.gz

Annotation:

ftp://ftp.ensembl.org/pub/release-101/gtf/ictidomys_tridecemlineatus/Ictidomys_tridecemlineatus.SpeTri2.0.101.gtf.gz

The gene annotation was filtered to include only protein-coding genes using the "cellranger mkgtf" module. 10X CellRanger was used to obtain transcript read counts for each cell barcode, filtered for cell barcodes called as cells based on the default parameters. Read count matrix was further processed using R 4.2.1, RStudio 2022.02.3, and Seurat 4.1.1[3]. Non-descriptive ground squirrel gene symbols (i.e. those starting with "ENSSTOG...") were replaced with gene symbols of mouse homolog genes, using the homolog conversion table from Ensembl. The suffix "[mh]" was appended to the mouse homolog genes to distinguish them from original ground squirrel gene symbols. The initial set of cells/barcodes was further filtered to include only those with ≥ 500 features/cells, ≥ UMIs/cells, and ≤ 10% of UMIs corresponding to mitochondrial genes (defined as those with the gene symbol starting with "MT-"). This resulted in ~11,000–20,000 cells/sample included in the dataset for further analysis, with the sequencing depth of ~70–100k reads/cell. Read counts were processed according to the standard Seurat analysis workflow, including normalization by "LogNormalize" method, identification of variable features, data scaling, PCA, clustering and visualization using UMAP plots. Violin and UMAP plots in figures report log-normalized gene expression values. All cells were initially clustered in an unbiased way using Seurat FindClusters function with a resolution parameter set to 0.5. Next, genes enriched in each cluster were determined using the FindAllMarkers function. The original clusters were assigned to cell types based on the manual analysis of the top marker genes. Clusters lacking common cell type markers among top enriched genes were assigned to the "Undefined" group. Clusters with top enriched genes corresponding to multiple cell types were assigned to the "Mixed" group. Neuronal cell type cluster was subset, independently re-processed, and further subclustered into 29 neuronal clusters. Top marker genes of each neuronal cluster were determined using the FindAllMarkers function. The cell abundance and top 3 markers of each neuronal cluster are shown in Extended Data Table 1. Statistical comparison of gene expression values between different conditions within specific cell populations was performed using the default non-parametric Wilcoxon rank sum test, as implemented in the FindMarkers Seurat function.

**De novo receptor cloning**

Total RNA was isolated from the arcuate nuclei of active and IBA animals that had been deeply anesthetized by isoflurane inhalation and subjected to intracardiac perfusion with ice cold PBS. The brain was rapidly dissected and a vibratome (Leica VT1200) was used to cut 300–600 µm coronal slices posterior to the separation of the optic chiasm. The area surrounding the third ventricle, including the arcuate nucleus and median eminence, were manually dissected out from the slices using 27 G needles and placed immediately into RNA lysis buffer from the Quick-RNA Microprep Kit (Zymo, R1050). Total RNA was isolated from tissue using the Quick-RNA Microprep Kit (Zymo, R1050). RNA concentration and integrity number (RIN) were assessed by an Agilent 2100 Bioanalyzer (Agilent, Santa Clara, CA). RNA concentrations were in the range of ~20 – 400 ng/µL and RIN values were in the range of 7.4–9.5. The resulting RNA was used for de novo cloning of *Ghsr* and long-form *Lepr*. cDNA was prepared (Invitrogen SuperScript III First-Strand Synthesis for RT-PCR, 18080-051) and the gene of interest amplified (Phusion High-Fidelity PCR Kit, E0553S) using the following primers for *Ghsr*: forward 5'- CCAACTTGATCCAGGCTCC −3', reverse 5'- CAAGTTCCGCTGTGCGATGG −3'; and *Lepr*: forward 5'- CAGGTACATGTCTCTGAAGTAAG −3', reverse 5'- GCCACGTGATC-CACTATAATAC −3'. Gel electrophoresis was used to isolate the band of interest and DNA extracted using the Qiagen Gel Extraction Kit (28704). ORFs were then ligated to topo vector (StrataClone Blunt PCR Cloning Kit, 240207). cDNA was sent for Sanger Sequencing (Genewiz), and reference sequences compared the NCBI database.

**Hypothalamic infusions**

Two hour and 24-h food consumption were measured in paired IBA animals in two separate experiments performed in the hibernaculum (4 °C, 40–60 % humidity, constant darkness) across two different hibernation seasons (Winter 2021: hibernation season 1 and Winter 2023: hibernation season 2) with two separate doses (15.3 nmol T3 for hibernation season 1, 15.3 pmol T3 for hibernation season 2).For hibernation season 1, core body temperature was measured by an abdominal implant, calibrated between 4 and 40 °C (EMKA Technologies, M1-TA), and interscapular temperature was measured by an interscapular implant (IPTT-300, BMDS). For hibernation season 2, only interscapular temperature (IPTT-300, BMDS) was used to assess body temperature. Animals were implanted during the active season, while they were euthermic, and allowed to recover for at least 2 weeks before implanted with hypothalamic infusion cannulas as described below.

In a subsequent surgery, infusion cannulas (10 mm 26 G guide, 11 mm 33 G internal, PlasticsOne) were implanted into the mediobasal hypothalamus during the active season while animals were euthermic. Briefly, animals were induced into, and maintained at, a stable anesthesia plane using isoflurane. Animals were administered 0.03 mg/kg preoperative buprenorphine subcutaneously. The scalp was shaved and the animal transferred to a stereotax (Kopf), where the skin was sterilized by repeated applications of betadine and 70% ethanol. Sterile technique was used to expose the skull and drill a hole to allow for cannula implantation. The following stereotaxic coordinates were utilized for cannula implantation: 0.5 mm posterior bregma, 0.8 mm lateral midline, 8 mm ventral (guide cannula)/ 9 mm ventral (infusion cannula). Two bone screws (2 mm long, 1.2 × 0.25 mm thread, McMaster-Carr) and dental cement (RelyX Unicem Resin, 3 M, 56830) were used to anchor the guide cannula to the skull. A dummy cannula was placed in the guide cannula until experiments were performed. Animals received a dose of 2 mg/kg meloxicam in 1.5 mL saline subcutaneously immediately after surgery. Animals received post-operative buprenorphine 0.03 mg/kg every 12 h and meloxicam 1 mg/kg every 24 h intraperitoneally for 48 h. Animals were allowed to recover for at least 2 weeks in pre-hibernation environmental conditions in the vivarium (20 °C, *ad libitum* food and water, 40–60% humidity, on Central Standard Time light-dark cycle) before they were brought to the hibernaculum (4 °C, no food or water, 40–60% humidity, constant darkness). Animals were given three days to enter torpor (body temperature <10 °C), after which they were monitored for at least 1 week to ensure regular IBA:torpor bouts. Animals that failed to enter or maintain hibernation during this time were excluded from the study and returned to the vivarium.

During hibernation season 1 (Winter 2021), feeding was first assessed at baseline (no injection, during IBA 3–4). Animals were allowed to return to torpor after control experiments. Hypothalamic T3 infusion (15.3 nmol per animal) was performed in the same animals after at least 1 subsequent IBA had elapsed (corresponding to IBA 5–7).

During hibernation season 2 (Winter 2023), feeding was first assessed at after control infusion (DMSO, during IBA 6–7). Animals were allowed to return to torpor after control experiments. Hypothalamic T3 infusion (15.3 pmol per animal) was performed in the same animals after at least 1 subsequent IBA had elapsed (corresponding to IBA 7–8), with the exception of one animal which was tested on subsequent IBAs. For both hibernation seasons, infusions were performed while animals were in the process of arousing from torpor. Animals were identified as IBA candidates when abdominal temperature exceeded 8 °C and/or interscapular temperature exceeded 10 °C. Squirrels were weighed and transferred to a clean cage in the hibernaculum, kept at 4 °C in constant darkness. For infusions, when abdominal temperature exceeded 16 °C and/or interscapular temperature exceeded 26 °C, a connector assembly consisting of PE50 tubing attached to an infusion cannula was loaded with control DMSO vehicle or T3 (Sigma, T2877) solubilized in DMSO (Sigma, D2650). At this point, animals were responsive to touch, but remained curled in the stereotypical torpor position and were unable to move. Infusion solution was dispensed in a 1 μL bolus at a rate of 0.33 μL/min. The infusion cannula was left in the guide cannula for two minutes to allow for the complete diffusion of the infusion solution. The infusion cannula was removed and replaced with a dummy cannula. For baseline (no infusion) experiments, animal body temperature was monitored until abdominal temperature exceeded 16 °C and/or interscapular temperature exceeded 26 °C.

After infusion or baseline monitoring, animals continued to warm up. Once the abdominal temperature surpassed 32 °C and/or the interscapular temperature surpassed 32 °C, animals became mobile and explored their cages. Animals were allowed to habituate for 20 min. Dog food was exclusively used for feeding consumption measurements. After habituation was complete, a pre-weighed amount of food was placed in the cage. Animals were allowed to feed for 2 h, at which point the remaining food was removed, weighed, and returned to the cage. Retrieving and weighing the food took less than 10 min per animal. The remaining food was returned to the cage and the animal allowed to feed for a further 22 h, to achieve a 24-h food consumption measurement.

## Hypothalamic tissue collection
Naïve animals that had not undergone any experiment were euthanized by isoflurane overdose and perfused with ice-cold PBS. The brain was removed from the skull and a ~ 6 mm thick section collected from the optic chiasm to the mamillary bodies using a rat coronal brain matrix (Electron Microscopy Sciences, 69083-C). The hypothalamus was isolated by removing brain matter above the top of the third ventricle and lateral to the optic tract. Tissue was flash-frozen in liquid nitrogen and stored at −80 °C until processing.

## Measurement of hypothalamic T3
Total triiodothyronine (T3) was extracted from frozen hypothalamus and purified as reported previously[4]. Briefly, hypothalamic tissue was homogenized in 100% methanol containing 1 mM 6-propyl-2-thiouracil (PTU) (Sigma, H34203) in a glass-glass tissue grind pestle (60 mm, Kontes, KT885300-0002). Homogenized tissue was centrifuged at 3000 rpm and supernatant removed. The pellet was resuspended and washed twice more in 100% methanol containing 1 mM PTU. T3 was extracted from supernatants and purified through solid-phase chromatography using 200–400 anion exchange chloride resin (Bio-Rad, 140-1251) in Poly-Prep chromatography columns (Bio-Rad, 731-1550). Columns were developed with 70% acetic acid (Spectrum, AC110) and washed twice with water. Supernatants were passed through the column without vacuum. T3 bound to columns was purified through a series of washes with acetate buffer pH 7.0 and 100% ethanol. T3 was eluted with 2.5 mL 70% acetic acid. Extracts were evaporated to dryness under nitrogen. T3 concentration was measured by ELISA (Leinco Technologies, T181). Dried product was resolubilized in the zero-standard and the kit run according to the manufacturer's instructions.

## Blood brain barrier tracer injections and analysis
Animals were anesthetized with isoflurane (4%) and injected in the tail artery with either biocytin-TMR (ThermoFisher, T12921) or 3 kDa dextran-TMR (ThermoFisher, D3307) at 10 mg/kg. Animals were allowed to recover in their home cage for 30 min until perfusion fixation with 4% paraformaldehyde as described for Immunohistochemistry.

Brains were sectioned on a Leica cryostat at 40 mm and every tenth section was imaged for blood brain barrier permeability analysis. Sections from dextran-injected animals were rinsed with PBS and coverslipped with Vectashield containing DAPI (Vector Labs, H-1200). For biocytin-injected animals, sections were immunostained to amplify fluorescence. After 1 h block with 10% bovine serum albumin (BSA) in PBS with 0.1% Triton-X-100, sections were incubated with Streptavidin-AlexaFluor594 (1:1000 in 0.1% PBS-TritonX-100, ThermoFisher S11227) for 2 h at RT. Sections were washed three times with PBS, then coverslipped with Vectashield as above. Z-stack images of liver and arcuate nucleus of the hypothalamus were acquired on a confocal microscope (Zeiss, LSM-780) using ZEN Software. Maximum intensity projection images were used for quantification in FIJI. Fluorescence intensity for the red channel was measured within circular ROIs manually drawn over ARC-ME, or the entire field of view for liver. Brain tracer fluorescence intensity was normalized to mean liver tracer fluorescence intensity from images with standardized acquisition settings.

## Statistics, analysis, and data collection
Statistical analyses were performed in GraphPad Prism v9.0 or higher (GraphPad Software, San Diego, CA) for all comparisons with the exception of sc-sequencing analysis, which was performed in R 4.2.1. Final figures were assembled in Adobe Illustrator. Data were tested for normality using the Shapiro-Wilk normality test. When normality was assumed, the Two-sided Student's t-test was used to compare two groups, One-Way ANOVA was used to compare multiple groups with one factor, and Two-Way ANOVA was used to compare multiple groups with two factors. Dunnett's multiple comparison was used to find *post hoc* differences with one factor. Tukey's multiple comparisons test was used to find *post hoc* differences among groups for Two-Way ANOVAs. Paired data were analyzed with a paired Two-sided Student's t-test. When data were not normal, the Two-sided Mann-Whitney test was used to compare two groups, and Two-Way ANOVA was used on rank transformed data to compare multiple groups with two factors that did not pass the Shapiro-Wilk normality test. Tukey's multiple comparisons test was used to find *post hoc* differences among non-normal groups with two factors. Statistical comparisons of gene expression data from single-cell RNA sequencing were performed using the Two-sided Wilcoxon rank sum test as implemented in the Seurat R package.

Sample sizes and statistical data are reported in the text and figure legends. In the text, values are provided as mean ± SEM, and $P < 0.05$ was considered statistically significant. No blinding was used for behavioral data collection. Immunohistochemistry quantification was performed blinded. Individuals in experimental groups were chosen to best match body weight and to represent both sexes across groups. As implemented in the Seurat R package.

## Reporting summary
Further information on research design is available in the Nature Portfolio Reporting Summary linked to this article.

## Data availability

All data are available in the main text or the supplementary materials. The RNA sequencing data was deposited to the Gene Expression Omnibus, accession number: GSE242381. Source data are provided with this paper.

## Code availability

No custom code was generated in this study.

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

## Acknowledgements

This work was funded by a Gruber Foundation Fellowship (SMM), The Kavli Foundation (MPP), National Institutes of Health grants R01DK126447, R01DK126447 (TLH), R01NS097547 (SNB), 1R01NS126271 (EOG), 1R01NS127058 (SNB and EOG) and National Science Foundation grants 1923127 (SNB) and 1754286 (EOG), 2323133 (EOG). RDP was supported by a scholarship from Coordenação de Aperfeiçoamento de Pessoal de Nível Superior (CAPES), Brazil. LV was funded by PID2021-125193OA-I00 by MCIN/AEI /10.13039/501100011033 and FEDER. We thank Lyle Murphy for technical assistance, and members of the Gracheva and Bagriantsev laboratories for comments and critique throughout the study. We thank Dr. Rachel Perry for performing the Leptin ELISA.

## Author contributions

Conceptualization: SMM, EOG, SNB. Data collection: SMM, RDP, MPP, VF, MS, LV, AK, HC, TLH, EOG. DKM supplied squirrels and provided advice on animal husbandry. Funding acquisition, project administration and supervision: EOG and SNB. Writing: SMM, RDP, EOG, SNB, TLH with contribution from DKM, MPP, HC.

## Competing interests

The authors declare no competing interests.
