## [Peer Review File · Nature Communications]

Hypothalamic hormone deficiency enables physiological anorexia in ground squirrels during hibernationEditorial Note: This manuscript has been previously reviewed at another journal that is not operating a transparent peer review scheme. This document only contains reviewer comments and rebuttal letters for versions considered at Nature Communications.

Reviewers' Comments:

Reviewer #1:

Remarks to the Author:

Mohr et al. do an excellent job at responding to many of the concerns raised in their initial submission, and should be commended for doing so with the noted technical challenges that are inherent to studying hibernation in ground squirrels. However, there are one major concern and one minor concern that are unresolved despite the author's responses.

The major concern is still that the concentration of T3 injected in the last figure, despite being decreased 1000x, is still supraphysiological. 15.3 pmol of T3 amounts to approximately 10 ng T3. 10 ng T3 infused into the squirrel hypothalamus (which we can estimate has a wet mass of 50-100 mg) would result in 100-200 pg/mg T3 infused into the hypothalamus. This is ~100-200 fold greater than active state animals, which is indeed supraphysiological. Moreover, the authors' citation of literature appears to point towards intraperitoneal injection of the same quantity of T3 which are very distinct from local intrahypothalamic injections. To make a stronger claim about the effects of T3 an infusion of the same order of magnitude as endogenous levels should be performed. We note that the authors may not be able to perform this experiment in a reasonable time period given the seasonality of the behavior, and thus an alternative would be to provide commentary to the readers that despite the fact that supraphysiological levels of T3 were used, the results point to or suggest that T3 has an effect on rescuing IBA anorexia. Moreover, the authors respond to the previous comment stating that the lack of full rescue of food intake by T3 suggests a more minor role for T3 in hibernation anorexia. The rebuttal to this comment argues that the effect size of food intake increases during T3 infusion (70% at 15.3 nmol and 120% at 15.3 pmol) is sufficient to match that of other papers, suggesting a significant physiological role. We, however, worry that a lack of concentration dependence to the effect suggests that either the injected amount is saturating even at the low concentration (see previous comment) or that the effect of rescue is not dependent on concentration. The statement: "Our data suggest that anorexia during hibernation is caused by hypothalamic T3 deficiency" is in our view too strong given the presented data, and should be reworded to something more similar to: "That supraphysiological T3 infusion is capable of opposing hibernation-induced anorexia, suggests the involvement of hypothalamic T3 deficiency in the overall effect of hibernation-induced anorexia." The minor concern: In the initial reviews, we requested a western blot of ghrelin receptor levels to confirm that levels are actually unchanged. The authors responded with the following: "While antibodies against squirrel ghrelin receptor are not available, this critique prompted us to perform additional experiments to quantify the level of Ghsr expression." A quick online search reveals antibodies against rat Ghsr, which are functional against mouse Ghsr as well. A BLAST between rat and squirrel Ghsr shows 97% sequence similarity, suggesting a high likelihood that the epitope is conserved. While there is a possibility that commercial rat Ghsr antibodies bind a different protein in squirrels, the researchers could attempt this experiment.

Finally, the title of the paper is a bit vague. The work is clearly about thirteen-lined ground squirrels, and thus should include some mention of the species, or at the very least hibernation, as this is the core focus of this paper.

Overall, this paper is clearly noteworthy in the field and we recommend it for publication under the

recommendation that the authors perform the requested validation experiments and/or lessen some of their claims, and revise the title.

Reviewer #3:

Remarks to the Author:

Mohr et al. have significantly improved the manuscript, adding in new experiments and analyses. They have addressed all concerns and I support the publication of this manuscript. The only small edit I suggest is in a slight overinterpretation as stated in the introduction - the authors state "Our experiments reveal that hibernation anorexia is caused by thyroid hormone deficiency in the hypothalamus." However, their experiments show that infusing T3 directly into the hypothalamus during IBA increased feeding over the next day. Thus the text should be amended to say that "Our experiments demonstrated that infusing T3 into the mediobasal hypothalamus during IBA increased feeding, which strongly suggests hibernation anorexia is caused by thyroid hormone deficiency in the hypothalamus."

We again thank the reviewers for their time and effort in reviewing our work, and for valuable suggestions.

Reviewer #1 (Remarks to the Author):

Mohr et al. do an excellent job at responding to many of the concerns raised in their initial submission, and should be commended for doing so with the noted technical challenges that are inherent to studying hibernation in ground squirrels. However, there are one major concern and one minor concern that are unresolved despite the author's responses.

The major concern is still that the concentration of T3 injected in the last figure, despite being decreased 1000x, is still supraphysiological. 15.3 pmol of T3 amounts to approximately 10 ng T3. 10 ng T3 infused into the squirrel hypothalamus (which we can estimate has a wet mass of 50-100 mg) would result in 100-200 pg/mg T3 infused into the hypothalamus. This is ~100-200 fold greater than active state animals, which is indeed supraphysiological. Moreover, the authors' citation of literature appears to point towards intraperitoneal injection of the same quantity of T3 which are very distinct from local intrahypothalamic injections. To make a stronger claim about the effects of T3 an infusion of the same order of magnitude as endogenous levels should be performed. We note that the authors may not be able to perform this experiment in a reasonable time period given the seasonality of the behavior, and thus an alternative would be to provide commentary to the readers that despite the fact that supraphysiological levels of T3 were used, the results point to or suggest that T3 has an effect on rescuing IBA anorexia.

As suggested, we added a "study limitations" paragraph that addresses this concern (below).

Moreover, the authors respond to the previous comment stating that the lack of full rescue of food intake by T3 suggests a more minor role for T3 in hibernation anorexia. The rebuttal to this comment argues that the effect size of food intake increases during T3 infusion (70% at 15.3 nmol and 120% at 15.3 pmol) is sufficient to match that of other papers, suggesting a significant physiological role. We, however, worry that a lack of concentration dependence to the effect suggests that either the injected amount is saturating even at the low concentration (see previous comment) or that the effect of rescue is not dependent on concentration. The statement: "Our data suggest that anorexia during hibernation is caused by hypothalamic T3 deficiency" is in our view too strong given the presented data, and should be reworded to something more similar to: "That supraphysiological T3 infusion is capable of opposing hibernation-induced anorexia, suggests the involvement of hypothalamic T3 deficiency in the overall effect of hibernation-induced anorexia."

As suggested, we added the following paragraph to Discussion:

193 "One limitation of our study is that a direct infusion of T3 into the hypothalamus creates a high local concentration of the hormone. Nevertheless, that supraphysiological T3 infusion is capable of opposing hibernation-induced anorexia, strongly supports the idea that long-term suppression of hunger during hibernation is, at least partially, due to central hypothyroidism."

We also toned down some of the statements in the Results section:

175 "Our data suggest that anorexia during hibernation could be caused by hypothalamic T3 deficiency."

183 "Thus, these data strongly suggest that the specific deficiency of T3 in the CNS contributes to reversible anorexia during hibernation."

The minor concern: In the initial reviews, we requested a western blot of ghrelin receptor levels to confirm that levels are actually unchanged. The authors responded with the following: "While antibodies against squirrel ghrelin receptor are not available, this critique prompted us to perform additional experiments to quantify the level of Ghnr expression." A quick online search reveals antibodies against rat Ghnr, which are

functional against mouse Ghnr as well. A BLAST between rat and squirrel Ghnr shows 97% sequence similarity, suggesting a high likelihood that the epitope is conserved. While there is a possibility that commercial rat Ghnr antibodies bind a different protein in squirrels, the researchers could attempt this experiment.

We agree with this concern. While we could not find an antibody against Ghnr that would recognize the squirrel protein, we also note that antibody staining would not fully address this concern regardless of the result, as the protein may be mislocalised, or has reduced downstream signaling. We acknowledged this possibility as follows:

197 “We showed, using single cell sequencing that the amount of *Ghnr* transcript does not decrease during hibernation. However, there remains a possibility that ghrelin receptor protein is functionally downregulated and thus may also contribute to seasonal anorexia.”

Finally, the title of the paper is a bit vague. The work is clearly about thirteen-lined ground squirrels, and thus should include some mention of the species, or at the very least hibernation, as this is the core focus of this paper.

We changed the title to “Hypothalamic hormone deficiency enables physiological anorexia during hibernation”.

Overall, this paper is clearly noteworthy in the field and we recommend it for publication under the recommendation that the authors perform the requested validation experiments and/or lessen some of their claims, and revise the title.

We thank the reviewer for the positive recommendation.

Reviewer #3 (Remarks to the Author):

Mohr et al. have significantly improved the manuscript, adding in new experiments and analyses. They have addressed all concerns and I support the publication of this manuscript. The only small edit I suggest is in a slight overinterpretation as stated in the introduction - the authors state “Our experiments reveal that hibernation anorexia is caused by thyroid hormone deficiency in the hypothalamus.” However, their experiments show that infusing T3 directly into the hypothalamus during IBA increased feeding over the next day. Thus the text should be amended to say that “Our experiments demonstrated that infusing T3 into the mediobasal hypothalamus during IBA increased feeding, which strongly suggests hibernation anorexia is caused by thyroid hormone deficiency in the hypothalamus.”

As suggested, we changed this sentence as follows:

58 “Our experiments show that infusing thyroid hormone into the mediobasal hypothalamus increased feeding during IBA, strongly suggesting that hibernation anorexia is caused by thyroid hormone deficiency in the hypothalamus.”

Reviewers' Comments:

Reviewer #1:

Remarks to the Author:

The authors have addressed our comments and we have no further comments. We may suggest that the authors include the species in the title since there are many hibernating animals and it is unknown whether mechanisms in one species translated to another. However, we leave this to the discretion of the authors. Congratulations on this study!